# Wheat amino acid transporters highly expressed in grain cells regulate amino acid accumulation in grain

Yongfang Wan[1], Yan Wang[1,2], Zhiqiang Shi[1,3], Doris Rentsch[4], Jane L. Ward[1], Kirsty Hassall[5], Caroline A. Sparks[1], Alison K. Huttly[1], Peter Buchner[1], Stephen Powers[5], Peter R. Shewry[1], Malcolm J. Hawkesford[1]*

1 Plant Sciences Department, Rothamsted Research, Harpenden, Hertfordshire, United Kingdom, 2 Triticeae Institute, Sichuan Agricultural University, Sichuan, P. R. China, 3 National Technology Innovation Center for Regional Wheat Production, Key Laboratory of Crop Physiology, and Ecology and Production in Southern China, Ministry of Agriculture, National Engineering and Technology Center for Information Agriculture, Nanjing Agricultural University, Nanjing, P. R. China, 4 University of Bern, Molecular Plant Physiology, Bern, Switzerland, 5 Computational and Analytical Sciences Department, Rothamsted Research, Harpenden, Hertfordshire, United Kingdom

* malcolm.hawkesford@rothamsted.ac.uk

**Data Availability Statement:** All the relavant data are in the supporting information S5 Table and the original SDS-PAGE image in S10 Fig.

## Abstract

Amino acids are delivered into developing wheat grains to support the accumulation of storage proteins in the starchy endosperm, and transporters play important roles in regulating this process. RNA-seq, RT-qPCR, and promoter-GUS assays showed that three amino acid transporters are differentially expressed in the endosperm transfer cells (*TaAAP2*), starchy endosperm cells (*TaAAP13*), and aleurone cells and embryo of the developing grain (*TaAAP21*), respectively. Yeast complementation revealed that all three transporters can transport a broad spectrum of amino acids. RNAi-mediated suppression of *TaAAP13* expression in the starchy endosperm did not reduce the total nitrogen content of the whole grain, but significantly altered the composition and distribution of metabolites in the starchy endosperm, with increasing concentrations of some amino acids (notably glutamine and glycine) from the outer to inner starchy endosperm cells compared with wild type. Overexpression of *TaAAP13* under the endosperm-specific HMW-GS (high molecular weight glutenin subunit) promoter significantly increased grain size, grain nitrogen concentration, and thousand grain weight, indicating that the sink strength for nitrogen transport was increased by manipulation of amino acid transporters. However, the total grain number was reduced, suggesting that source nitrogen remobilized from leaves is a limiting factor for productivity. Therefore, simultaneously increasing loading of amino acids into the phloem and delivery to the spike would be required to increase protein content while maintaining grain yield.

## Introduction

Wheat contributes about 10% of the dietary intake of protein in the UK, and grain protein content (GPC) is a key contributor to breadmaking quality. Farmers routinely increase the

**Funding:** Rothamsted Research receives grant-aided support from the Biotechnology and Biological Sciences Research Council (BBSRC) for the Designing Future Wheat strategic program (BB/P016855/1). Funder did not play any role in the study design, data collection and analysis, decision to publish, or preparation of the manuscript.

**Competing interests:** The authors have declared that no competing interests exist.

grain protein content by applying more inorganic nitrogen fertilizers, which increases the cost of production and may have an adverse environmental footprint. Therefore, improved nitrogen use efficiency is a long-term strategy for sustainable improvement of wheat productivity and grain protein content.

Amino acids are the major transported form of reduced nitrogen in the plant [1, 2]. The transport of amino acids across membranes and translocation from source to sink is mediated by membrane transport proteins: amino acid transporters (AATs). These are classified into two major subfamilies: the Amino Acid/Auxin Permease family (AAAP) and the Amino acid-Polyamine-Choline transporters family (APC) [3, 4]. In addition, a new group, the Usually Multiple Amino Acids Move In and Out Transporter (UMAMIT) family, has been identified [5], which is part of the Drug/Metabolite Transporter (DMT) family [6]. More than 110 AAT genes have been identified in *Arabidopsis* and genome-wide surveys of amino acid transporters have been reported for many plant species including rice [7], poplar [8], potato [9] and wheat [10].

Only a few AAT genes have been functionally characterized, mainly in *Arabidopsis*. Nevertheless, studies indicate that they play important roles in amino acid uptake into roots, phloem loading, long-distance transport and loading into the seed [11, 12]. Crucial functions for the import of amino acids into sink tissues (seed, fruit, and tuber) have been described for different amino acid transporters. The *AtAAP1* transporter is highly expressed in the embryo epithelium (transfer cells) of *Arabidopsis* and involved in uptake and transport of amino acids from the endosperm into the embryo, with *ataap*1 mutants having lower seed protein content and accumulation of high level of amino acids in seeds [13]. RNAi-mediated down-regulation of the *StAAP1* transporter can reduce the levels of free amino acids by 50% in potato tubers [14]. The tonoplast-localized *UMAMIT24* of *Arabidopsis* is able to transport the amino acids temporarily stored in the vacuoles of chalaza cells before being delivered to filial tissue, while the plasma membrane-localized *UMAMIT25* is expressed in the endosperm and pericarp and could mediate amino acid export from the endosperm [15]. The *UMAMIT18* transporter is present in vascular tissues and developing seeds, supporting accumulation nitrogen in developing siliques [16]. *UMAMIT11* and *14* AATs are expressed in the chalaza (unloading domain) of developing seeds and *UMANIT28* and *29* transporters export amino acids from the endosperm and pericarp, respectively. Single loss-of-function mutants of these four transporters resulted in the accumulation of high levels of free amino acids in the seeds and greatly reduced seed size [17]. By contrast, overexpression of *VfAAP1* (which is highly expressed in embryonic storage parenchyma cells at early maturation) in both pea and *Vicia narbonensis* (Narbon bean) seeds resulted in increased seed protein content (by 10–25%) and seed size (by 20–30%), by increasing import of amino acids into the embryo [18]. Similarly, the simultaneous overexpression of *PsAAP1* in the phloem and embryos of pea plants increased seed yield and seed storage protein levels when the plants were grown with highly abundant N, due to increased source-to-sink allocation of amino acids [19] and increased nitrogen use efficiency [20]. The *OsAAP6* transporter, which is highly expressed in the endosperm of rice, functions as a positive regulator of GPC and overexpression in rice greatly increased GPC in rice grain when overexpressed [21]. Tomato *SlCAT9* is a tonoplast Glu/Asp/GABA transporter, and overexpression of *SlCAT9* greatly affects the flavor profile of the tomato fruit by increasing the accumulation of these amino acids during fruit development [22].

Amino acid transporters not only affect grain nitrogen accumulation, but may also contribute to grain yield by changing the distribution of nitrogen between different tissues. *AtAAP8* played an important role in translocation of amino acids from source leaf phloem to sink embryo, and *ataap8* mutants reduced seed number by about 50% [23, 24]. Down-regulation of the expression of *OsAAP3* and *OsAAP5* or overexpression of *OsAAP1* increased rice grain

yield by increasing tiller number resulting from promoting axillary bud outgrowth [25–27]. Altered xylem-phloem transfer of amino acids in the *ataap2* mutant also increased seed yield and oil content in *Arabidopsis* by changing the source-sink translocation of amino acids [28].

Wheat grains comprise three distinct parts, the embryo, the outer layers (nucellar epidermis, testa and pericarp), and the endosperm, which account for about 3%, 7%, and 90% of the grain weight, respectively [29]. The endosperm consists of three cell types, endosperm transfer cells (ETC), aleurone cells (AL) and starchy endosperm cells (SE). Amino acids are unloaded from the phloem via the vascular bundle into the endosperm cavity of the grain [30], where they are actively taken up by transfer cells; this is the first bottleneck for nutrient entry into the endosperm. Subsequently, amino acids are transported into the starchy endosperm where they are utilized for protein synthesis [31]. The epithelium of the scutellum also takes up nutrients from the apoplast to support embryo development and protein synthesis. Therefore, the amino acid transporters in these grain cells may play crucial roles in regulating nitrogen accumulation in wheat grains.

Our previous work identified three amino acid transporters (*TaAAP2*, *TaAAP13*, and *TaAAP21*) which are highly expressed in different grain cells [10]. In this study, their expression patterns and roles in amino acid transport and nitrogen accumulation in wheat grains were characterized by heterologous expression and transgenesis.

## Materials and methods

### Expression analysis by RNA seq.

Expression data for three amino acid transporters were extracted from RNA-seq data [10, 32, 33]. Expression units are in FPKM (frequency per kilobase million).

### Harvest of materials and RNA extraction

The wheat cultivar Hereward was grown in field trials at Rothamsted Research in 2015, with either 200 kg N/ha fertilizer or no nitrogen application. For the 200 kg N/ha application, 50, 100, and 50 kg N/ha nitrogen (as ammonium nitrate) were applied at tillering, stem extension, and flag leaf emergence stages, respectively. Whole caryopses were harvested at 5, 10, 14, 17, 21, and 28 DPA (days post anthesis), and roots and leaves and stems at Zadoks 23 (2–3 tillers stage), Zadoks 45 (booting stage), and 14 DPA and stored at −80°C for subsequent RNA extraction and real-time PCR.

RNA extraction was modified as described by Chang et al. [34]. Frozen tissues were ground in liquid nitrogen and extracted in CTAB (cetyl trimethylammonium bromide) buffer (2% (w/v) CTAB, 2% (w/v) PVP K30, 100 mM Tris-HCl, pH 8.0, 25 mM EDTA, 2.0 M NaCl, 0.5 g/l spermidine, 2% (w/v) β-mercaptoethanol). The supernatant was extracted twice with chloroform:IAA (24:1) to remove proteins. RNA was precipitated by addition of 0.25 volumes of 10 M LiCl and incubation on ice overnight. The RNA pellet was dissolved in SSTE buffer (1.0 M NaCl, 0.5% (w/v) SDS, 10 mM Tris HCl pH 8.0, 1 mM EDTA) to remove polysaccharides and extracted once with chloroform:IAA. After ethanol precipitation, total RNA was dissolved in DEPC-treated water and stored at -80°C.

Total RNA was treated with DNase to remove genomic DNA contamination and purified through RNeasy mini spin columns (Qiagen). Two μg of RNA was used for reverse transcription with SuperScriptTM III reverse transcriptase (Invitrogen) to synthesize cDNA using anchored oligo (dT) 23 primers (Sigma-Aldrich).

## Real time PCR

Real time PCR was performed using an ABI7500 (Applied Biosystems) thermocycler. cDNA diluted 1:5 was used for RT–qPCR in a 25 μl reaction with SYBR ® Green JumpStart ™ Taq ReadyMix (Sigma-Aldrich).

The Cell division control protein AAA-superfamily ATPases (TraesCS4A02G035500) was used as an internal control gene as it showed the most stable expression across different wheat tissues and developmental stages [35]. The primers designed for RT-qPCR are shown in S1 Table. For each pair of primers, PCR efficiency was calculated in each run from a pool of all available cDNAs by using the LinRegPCR software [36]. All time points had three biological replicates. Normalised relative quantity (NRQ) was calculated by CT values and primer efficiency (E) of the target gene (T) in relation to the internal control gene (N) following formula: $NRQ = 1000X (E_{(T)})^{-CT,T} / (E_{(N)})^{-CT,N}$ [37].

## Plasmid vector construction

For promoter::GUS constructs, 1.412, 1.582 and 1.178 kb promoter fragments of *TaAAP2B* (B-genome), *TaAAP13D* (D-genome), and *TaAAP21A* (A-genome) based on their expression levels and the length of the promoter sequence available in the database were amplified from wheat leaf genomic DNA and cloned into pGEM-T Easy vector (Sigma), and then subcloned into vector pRRes104.293 containing β-glucuronidase (GUS) with PmeI and NcoI restriction sites. For the RNAi plasmid construct of *TaAAP13*, a 538bp fragment (125 to 662bp from ATG) of the coding region was cloned into pGEM-T Easy vector, then subcloned into the RNAi cassette in pRRes104RR.132 which contains with two pairs of restriction enzyme sites of BgIII/BsrGI and BamHI/BsiWI surrounding the maize ADH2 (alcohol dehydrogenase) intron. Expression is driven by the maize ubiquitin promoter plus intron and Nos terminator. For overexpression lines of *TaAAP13*, the full length of coding region was subcloned into vector pRRes104RR.161 under the control of the wheat endosperm-specific 1Dx5 HMW-GS promoter or vector pRRes104RR.125 under the control of the constitutive maize ubiquitin promoter plus intron and Nos terminator.

## GUS assay

Fresh grains at 7, 14, 21, 28 and 33 DPA were cut transversely or longitudinally and incubated in staining buffer (1 mM X-Gluc, 100 mM sodium phosphate pH 7.0, 0.5 mM potassium ferrocyanide, 0.5 mM potassium ferricyanide and 2% Triton X-100) at 37˚C for two hours to overnight. Roots, leaves and stems were vacuum-infiltrated for 5 minutes before incubation in staining buffer. GUS-stained tissues were visualized directly or after de-staining with ethanol and photographed using a Leica MA250 camera.

## Wheat transformation

The wheat cultivar Cadenza was transformed using particle bombardment to deliver plasmid DNAs into immature embryos and transgenic plants regenerated via somatic embryogenesis [38]. Transgenic plants were confirmed by PCR using gene-specific primers. Transformed plants were grown to maturity in a containment glasshouse.

## Homozygous assay and growth of transgenic plants

DNA was extracted from leaves of T1 seedlings and used for determination of homozygosity and transgene copy numbers by iDna Genetics Ltd (Norwich Research Park, UK). Wheat plants were grown in a GM glasshouse with 16 hours day at 20˚C and 8 hours night at 15˚C.

The plants were watered once a day using a flood bench system lined with capillary matting. Four biological replicates comprising 4 plants each were grown in a randomized order.

## Complementation of yeast amino acid transporter mutants

Full length coding regions of *TaAAP2*, *TaAAP13* and *TaAAP21* were PCR amplified from cDNA of 21 DPA wheat grains, cloned into pGEM-T Easy vector and confirmed by sequencing. The three AAP genes were subcloned into vector PDR196 between a PMA (plasma membrane H+-ATPase) promoter and ADH terminator, and *Saccharomyces cerevisiae* strains 22574d (Matα, ura3-1, gap1-1, put4-1, uga4-1) [39], 22Δ6AAL (Matα, ura3-1, gap1-1, put4-1, uga4-1, can1::hisG, lyp1/alp1::hisG, lys2::hisG) [40], 21.983c (MATa, gap1-1, can1-1, ura3) [41], YDR544.137 (MATα, ura3-1, gap1-1, put4-1, uga4-1, ssy1::kanMX, dip5::hisG, agp1:: hisG, gnp1::kanMX*)*, and 30.537a (Matα, gap1-1, dip5::kanMX2, ura3) were transformed according to the method described previously [42]. The transformants, negative control (empty vector) and positive controls (containing *Arabidopsis* AAPs) were selected on nitrogen-free media containing 1g/l for proline, glutamate and GABA, 1mM for citrulline, glycine, lsoleucine, methionine, phenylalanine, and valine, 5mM for leucine, threonine, tryptophan and tyrosine, 0.5g/l for arginine, and 0.1mM for lysine with 1g/l Urea. For non-selective conditions, media were supplemented with 0.5g/l ammonium sulfate and grown at 30°C for 5–7 days [43].

## Nitrogen determination

Total nitrogen was determined using the American Society for Testing and Materials (ASTM) standard protocol E1019 using a Leco combustion analysis system based on the Dumas method. 150 mg of wholemeal wheat grain flour and four biological replicates were analyzed.

## Pearling of grains

20g mature seeds were pearled in a Streckel & Schrader (Hamburg, Germany) pearling mill [44]. Five fractions were prepared by sequential pearling, corresponding to about 4, 7, 7, 12 and 10% of the grain weight, and the remaining cores (60%) and whole grains were milled in a centrifugal mill (Retsch, ZM200). The pearled fractions are enriched in embryo and pericarp tissue (F1), aleurone layer (F2), sub-aleurone layer (F3) and two progressively more central areas of the starchy endosperm (F4 and F5), respectively. Three biological replicates were performed.

## Grain area, length, width, moisture and biomass measurement

Seed area, length and width were measured using a MARVIN- Digital Seed Analyzer SN 176 (Marvitech—Germany). Four biological replicates from 16 plants (200–400 grains from each biological replicate) were analyzed. Seed moisture was determined using a Bruker Minispec mq20 TD-NMR Contrast Agent analyzer (Germany) with bespoke robot and recording software (ROHASYS, Dutch Robotics) by loading on samples of 7-10g seeds. Thousand grain weight (TGW) was determined and expressed on a 15% seed moisture grain weight. Above ground biomass (stems and leaves) was determined after oven drying at 80°C overnight.

## SDS–PAGE

10mg flour was suspended in 300μl total protein extraction buffer [50 mM Tris–HCl, pH 6.8, 2% (w/v) SDS, 10% (v/v) glycerol, 2% (w/v) dithiothreitol (DTT) and 0.1% (w/v) bromophenol blue] [45]. The extracts were heated for 3 minutes at 95°C, and centrifuged for 15 minutes. 3 μl

of the supernatants were separated on pre-cast 4–12% Bis-Tris Nu-PAGE gels (Invitrogen). The four biological extracts, and two technical gel replicates were performed. The gels were stained with Coomassie BBR250 in 10% (w/v) trichloroacetic acid (TCA), 40% (v/v) methanol, and de-stained in 10% (w/v) TCA. The gels were scanned with a HPG4010 scanner, and the images from grey tif files were processed with Phoretix 1D advanced software (Nonlinear Dynamics, Durham, NC, USA). The proportions of protein groups were expressed as a % of total gluten proteins.

## [1]H-Nuclear magnetic resonance (NMR) spectroscopy

Sample preparation for [1]H-NMR was carried out according to the method previously described [46]. Wholemeal samples (30 mg) (three technical replicates each of three biological replicates) were extracted in 80:20 $D_2O:CD_3OD$ containing 0.05% d4–trimethylsilylpropionate (TSP; 1ml) as an internal standard at 50°C for 10 min. After centrifugation (5 min at 13 000 rpm), the supernatant was removed and heated to 90°C for 2 min to halt enzyme activity. After cooling and further centrifugation, the supernatant (650 μl) was transferred to a 5 mm NMR tube for analysis. The data collection and analysis were described as in this method [47].

## High performance liquid chromatography (HPLC)

Free amino acids were extracted according to the method previously described [48]. 30 mg samples (four biological replicates) were suspended in 500 μl of 0.01 N HCl by shaking for 30 min at room temperature. After centrifugation at 10000 rpm for 15 minutes, the supernatants were filtered through a 0.45 μm poly (ether sulfone) filter before analysis. Amino acids were separated using a Waters Alliance 2795 HPLC system (Waters Corp., Milford, USA) coupled with a Waters 474 scanning fluorescence detector. The detailed method is described as previously reported [49].

## Wheat grain fixation, embedding and light microscopy

The middle parts of 3 mature grains were cut into 2 mm transverse sections and immediately fixed in 4% (w/v) paraformaldehyde in 0.1 M Sorenson's phosphate buffer ($NaH_2PO_4.2H_2O$ and $Na_2HPO_4. 12H_2O$, pH 7.0) with 2.5% (w/v) glutaraldehyde overnight. After dehydration in increasing concentrations of ethanol, the sections were embedded in LR White Resin for two weeks at room temperature and polymerized at 55°C. A Reichert-Jung Ultracut ultramicrotome was used to section the resin-embedded grains at 0.5 μm or 1 μm thickness for protein staining. Protein bodies were stained with 1% (w/v) Naphthol Blue Black in 7% (w/v) acetic acid. The slides were visualized under a Zeiss Axiophot microscope and images were acquired with a RetigaExi CCD digital camera (Qimaging, Surrey, BC, Canada) under bright field optics and MetaMorph software version 7.5.5 9 (Molecular Devices, Sunnyvale, CA, USA).

## Statistical analysis

Data were analyzed using ANOVA accounting for the randomized block design. Where necessary, data were log or square root transformed to satisfy homogeneity of variance. The least significant difference (LSD) values presented are the LSD associated with comparisons. Analyses were carried out using the GenStat (19th edition, VSN International Ltd., Hemel Hempstead, U.K.). Principal component analysis (PCA) and partial least squares discriminant analysis (PLS-DA) were applied using the correlation matrix between variables. Each input variable was transformed according to the univariate analysis, adjusted for any block effects and averaged over technical replicates. PCA was performed in Genstat, 19th edition. The

software Simca-P v. 16 (Sartorius Stedim Data Analytics AB) was used for OPLS-DA. The analysis was carried out using quantified 1H-NMR data scaled to unit variance.

## Results

### The amino acid transporters *TaAAP2*, *TaAAP13*, and *TaAAP21* are differentially expressed in wheat grain cells and plant organs

Previously three *TaAAP* genes (*TaAAP2*, *TaAAP13*, and *TaAAP21*) out of 100 homoeologous groups of amino acid transporters (283 genes) were shown to be highly expressed in the endosperm transfer cells, starchy endosperm cells and aleurone cells of wheat grains, respectively [10]. These genes were therefore selected to determine their gene expression patterns, spatial tissue localization and potential functions in grain nitrogen metabolism.

The expression patterns determined from RNA-seq data of endosperm transfer cells, starchy endosperm cells, and aleurone cells at 20 DPA (days post anthesis) [10, 50] are shown in Fig 1A–1C. *TaAAP2* (three homoeologs with the IWGSC RefSeq v1.1 IDs: TraesCS2A02G348600 (*TaAAP2A*), TraesCS2B02G367000 (*TaAAP2B*), TraesCS2D02G347000 (*TaAAP2D*)) was highly expressed in endosperm transfer cells at 20 DPA (corresponding to the middle of grain filling), but had very low expression in starchy endosperm and aleurone cells at 20 DPA (Fig 1A). *TaAAP13* (TraesCS4A02G215300 (*TaAAP13A*), TraesCS4B02G100800 (*TaAAP13B*), TraesCS4D02G097400 (*TaAAP13D*)) was highly expressed in starchy endosperm and aleurone cells (Fig 1B) while *TaAAP21* (TraesCS7A02G356639 (*TaAAP21A*), TraesCS7B02G271151 (*TaAAP21B*), TraesCS7D02G366000 (*TaAAP21D*)) was more highly expressed in aleurone cells compared with starchy endosperm and transfer cells (Fig 1C). RNA-seq showed that the three AAP genes are more highly expressed in developing grains than vegetative organs prior to anthesis [33] (S1 Fig). Gene expression profiles were therefore determined in further stages of grain development and in vegetative organs of field-grown wheat at two nitrogen levels (with fertilizer applied at 0 kg N/ha, and 200 kg N/ha), using RT-qPCR (Fig 1D–1F). *TaAAP2* and *TaAAP13* were highly expressed during the middle stages of grain filling (14 and 21 DPA), while the expression of *TaAAP21* increased during the late stage of grain filling (28 DPA). Nitrogen treatment did not greatly affect the expression levels of *TaAAP2* and *TaAAP13* in grains, but negatively affected their expression in vegetative organs at some growth stages. The gene expression patterns determined by RT-qPCR generally agreed with the RNA-seq data.

### Localization of *TaAAP2*, *TaAAP13* and *TaAAP21* expression by promoter::GUS expression

Promoter::β-glucuronidase (GUS) transgenic lines were produced in order to determine precise gene expression patterns. GUS expression was determined in T1 grains of 10 lines and in T2 grains of one (*TaAAP2B* and *TaAAP21A*) or two lines (*TaAAP13D*). These gave similar results which mostly confirmed the expression patterns shown by RNA-seq and RT-qPCR (Fig 2).

*TaAAP2B* promoter::GUS expression was observed in transfer cells from 7–28 DPA (Fig 2), with the strongest expression at 21 DPA, indicating that *TaAAP2* is transfer cell-specific and may play a role in loading amino acids into the starchy endosperm during grain filling. GUS staining was observed in the lobes of the starchy endosperm of lines expressing the *TaAAP13D* promoter::GUS during early grain development (7 DPA), but the staining was faint and required overnight incubation (Fig 2). However, it was highly expressed in the starchy endosperm from 14 to 28 DPA and clearly seen after two hours staining. A clear gradient in intensity of staining from the sub-aleurone cells to the inner starchy endosperm was observed at 14

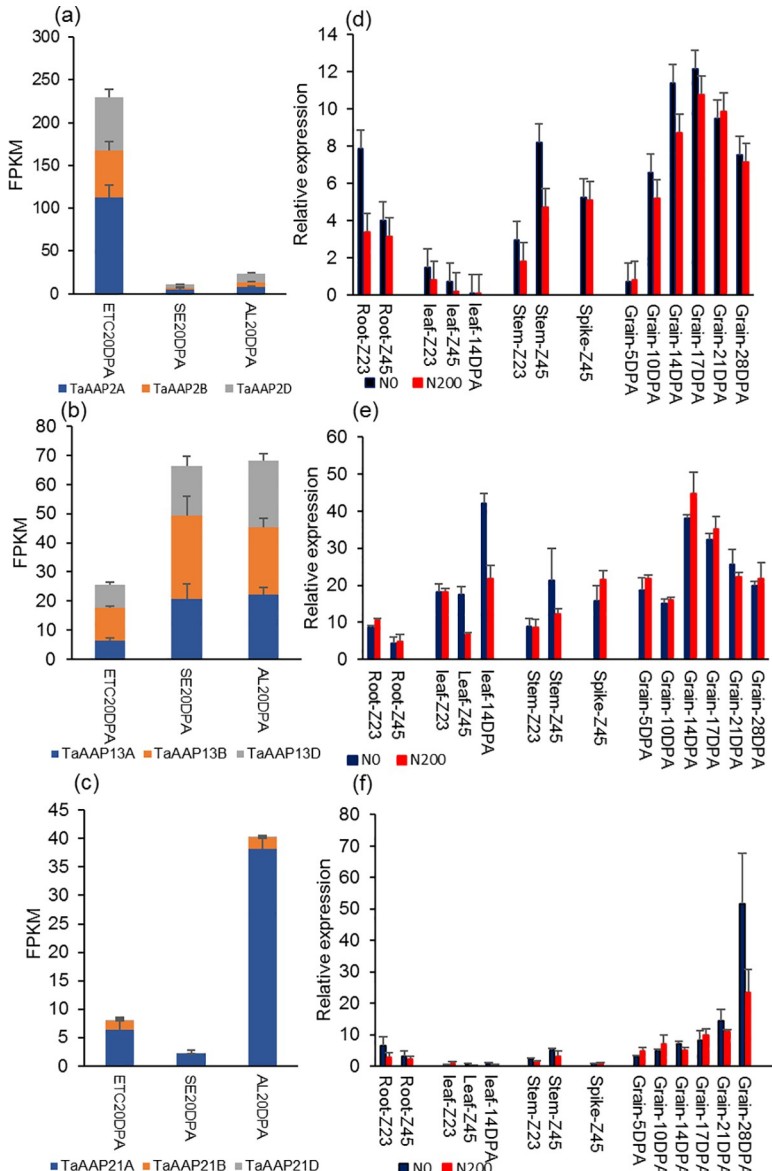

**Fig 1. The expression patterns of *TaAAP2*, *TaAAP13*, and *TaAAP21*.** (a-c), Expression of *TaAAP2* (a), *TaAAP13* (b), and *TaAAP21* (c) in grain cells: endosperm transfer cells (ETC), starchy endosperm (SE), aleurone cells (AL) of Chinese Spring at 20DPA (days post anthesis) by RNA-seq data [10, 32]. (d-f), Expression of *TaAAP2B* (d), *TaAAP13D* (e), and *TaAAP21A* (f) in different organs of wheat cultivar Hereward from whole caryopses at 5, 10, 14, 17, 21, and 28 DPA (days post anthesis), roots, leaves and stems at Zadoks 23 (2–3 tillers stage) and Zadoks 45 (booting stage) at two nitrogen levels, 0kg N/ha and 200kg N/ha, by RT-qPCR. Error bars represent standard errors (SE).

DPA with the strongest expression being in the sub-aleurone cells at 21 to 28 DPA (Fig 2, S2A and S2B Fig). The inner starchy endosperm cells did not exhibit any GUS staining even after overnight incubation. No expression was observed in the aleurone cells (Fig 2, S2A Fig), demonstrating that *TaAAP13* was specific for the starchy endosperm, and suggesting that the high expression level in the aleurone cells shown by RNA-seq resulted from contamination during tissue preparation [50]. Similarly, no GUS staining was observed in the embryos of *TaAAP13D* promoter::GUS transgenic lines. The *TaAAP13D* GUS expression patterns imply that it may function in transporting amino acid across the starchy endosperm.

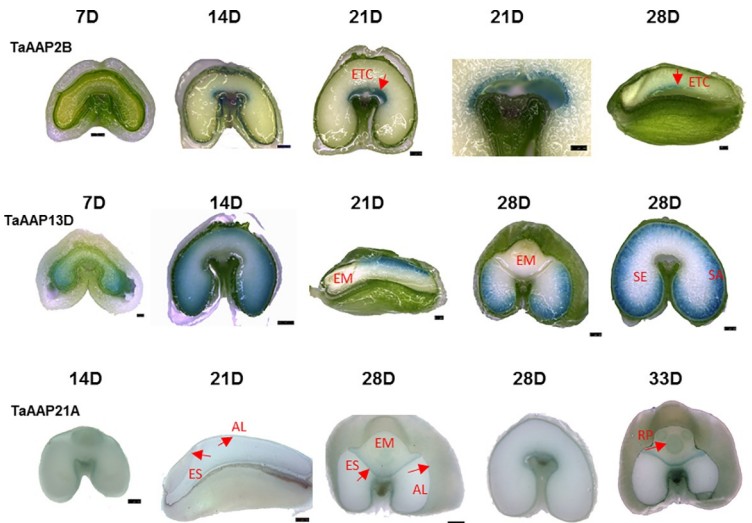

**Fig 2. Localization of *TaAAP2B*, *TaAAP13D*, and *TaAAP21A* by promoter::GUS expression.** The GUS expression patterns driven by promoters of *TaAAP2B*, *TaAAP13D*, and *TaAAP21A* in T2 developing grains at 7, 14, 21, 28 and 33DPA (days post anthesis). The GUS staining was visualised directly for *TaAAP2B* and *TaAAP13D* or after serial ethanol de-staining for *TaAAP21A* under stereomicroscope Leica MA250. Scale bars represent 500 μm except *TaAAP2B* (21D enlarged) and *TaAAP13D* (7D) for 250 μm. ETC: endosperm transfer cells, EM: embryo, AL: aleurone cells, ES: epithelium of scutellum, SE: starchy endosperm, RP: root primordia.

*TaAAP21A* promoter::GUS activity was detected in the epithelium of the scutellum, the aleurone, and the transfer cells (Fig 2, S2C–S2F Fig). GUS staining was very weak at 14 DPA (Fig 2), but increased in both, the aleurone and the epithelium of the scutellum from 21 to 28 DPA (Fig 2, S2C–S2F Fig), which is consistent with the data from RT-qPCR and RNA-seq. However, *TaAAP21A* was much more strongly expressed in the epithelium of the scutellum and root primordia, but less in the aleurone at 33 DPA. The expression pattern of *TaAAP21A* indicates that *TaAAP21* may transport amino acids into aleurone cells for storage protein synthesis, and take up amino acids into the scutellar epithelium for the embryo development.

No GUS staining with the three promoter::GUS constructs was observed in leaf, root or stem tissues of plants grown in the glasshouse (well-watered with high nitrogen) over two generations. Very weak GUS expression was detected in the tips of the roots in germinating seeds after 2–3 days imbibition in water (S2G–S2I Fig).

## The amino acid transporters are able to transport broad ranges of amino acids in yeast mutants

Yeast (*Saccharomyces cerevisiae*) has been used as a heterologous expression system to characterize many plant transporters, using mutant strains that lack transporters for specific essential components such as amino acids. In order to functionally characterize and determine the selectivity of *TaAAP2*, *TaAAP13*, and *TaAAP21*, yeast mutants lacking transporters for 16 endogenous amino acids were transformed with plasmids containing the full-length gene coding regions of the three wheat amino acid transporters and growth of transformants was tested on media containing different amino acids as the sole nitrogen sources or as the sole source for lysine. This showed that all three wheat transporters can transport a broad spectrum of amino acids, some of which are shared (S2 Table, and S3 Fig). Most of amino acids transported by the three amino acid transporters are neutral amino acids (Pro, Gln, Gly, Leu, Ile, Met, Phe, Val, Thr, Trp, and Tyr), but acidic (Glu) and/or basic (Arg and Lys) amino acids are also

transported. *TaAAP2* can transport a wide range of amino acids, particularly uncharged amino acids, but cannot transport Gln, Gly (uncharged), Glu (acidic), Lys or Arg (basic). By contrast, *TaAAP13* and *TaAAP21* can transport Gln, which is the major transported amino acid in plants, Glu and other neutral, basic and acidic amino acids.

## Functional analysis of amino acid transporter *TaAAP13* by RNAi-suppression

TaAAP13 is highly expressed in the starchy endosperm and hence may provide amino acid substrates for storage protein synthesis in this tissue. Transgenic RNA interference (RNAi) lines were therefore generated to explore its role in more detail. The RNA expression levels in T3 grains of three *TaAAP13 RNAi* lines (SE-1R, 9R, 10R) at 14 DPA (the stage of peak expression) were reduced 44–70% compared with null lines from the same transformation events (SE-3N, SE-11N) and with a non-transgenic control line (wild type, SE-24WT) (Fig 3A). There are no significant differences between RNAi and null lines in grain numbers, thousand grain weight (TGW), biomass, grain yield per plant and grain areas except that SE-10R grain had a greater area than the SE-11N grain (S4 Fig).

To determine whether the accumulation of total nitrogen in the grains was reduced, the nitrogen concentrations in wholemeal flour and white (starchy endosperm) flour of RNAi lines were compared with those of the null lines. Although the concentrations of nitrogen in the RNAi lines were slightly lower than those in the nulls, the differences were not statistically significant (Fig 3B). To determine if the nitrogen gradient across the grain was altered, five pearling fractions (F1, F2, F3, F4, and F5, enriched in the bran, aleurone, sub-aleurone, outer starchy endosperm and inner starchy endosperm respectively) and the remaining cores of two RNAi (SE-1R, SE-10R) lines were compared with null and wild type lines. The pearling fractions corresponded to 4, 7, 7, 12 and 10% of the grain weight, and the remaining cores to 60% of the total grain weights of the lines. No significant differences in the concentrations of nitrogen in the fractions were observed between the two RNAi lines (SE-1R and SE-10R), or between the RNAi lines and the null (SE-3N) and non-transgenic control (SE-24WT) lines (S5A Fig). However, the total concentration of free amino acids determined by HPLC was

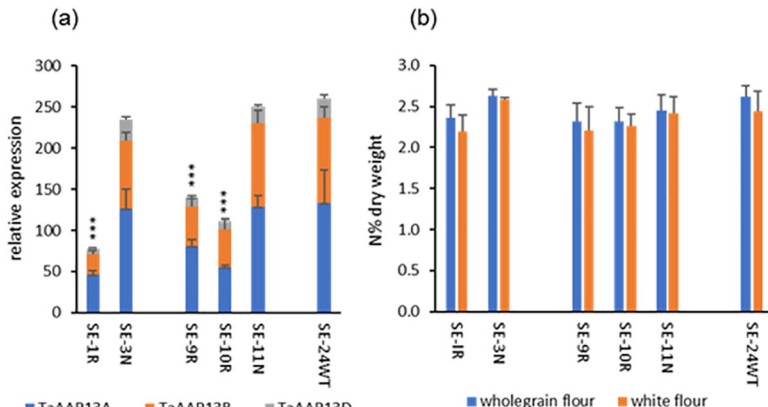

**Fig 3. *TaAAP13* expression and nitrogen concentrations in RNAi lines (SE-1R, SE-9R, SE-10R), null lines (SE-3N, SE-11N), and non-transgenic line SE-24WT (Cadenza, wild type).** (a), *TaAAP13* gene expression at 14DPA of wheat developing grain. Asterisks (\*\*\*) indicated the significant differences between RNAi lines and nulls or wild type detected by ANOVA (P<0.001, F-test). Error bars represent standard errors. (b), Nitrogen concentrations of wholemeal flour and white flour. The differences between RNAi lines and nulls were not statistically significant (P<0.05, F-test).

higher in RNAi line SE-1R (S5B Fig), which had the highest level of RNA suppression (Fig 3A).

Wholemeal and pearling samples from RNAi SE-1R and the corresponding null line (SE-3N) were therefore analyzed for polar metabolites (which include 11 amino acids) using $^1$H-NMR spectroscopy (S3A and S3B Table). Supervised multivariate analysis (OPLS-DA) clearly separated the wholemeal flour from the SE-1 and SE-3 lines (Fig 4A), with the scores contribution (Fig 4B) showing that the majority of metabolites were significantly elevated in SE-1R ($P<0.05$, F-test): glutamine by 87%, asparagine by 57%, glycine by 50%, maltose by 279%, and glucose by 105% (S3A and S3B Table). The compositions of the pearling fractions from the two lines were compared by PCA (Fig 4C). The different fractions are separated in PC1, which accounts for 75% of the total variation, while the two lines are separated in PC2, which accounts for 14% of the total variation. The loading plot for PC1 (Fig 4D) shows that the concentrations of all metabolites decreased from the bran (F1) to the core in both lines (SE-1 and SE-3), except for maltose which increased from F1 to F5 (S5C Fig, S3A and S3B Table). The loading plot for PC2 shows that amino acids accumulated to higher concentrations in SE-1R than in SE-3N, with glutamine, aspartate, and glycine being increased by 16%-52%, 29.9–41.8%, 21–62%, respectively, in the different fractions. The results therefore demonstrated that

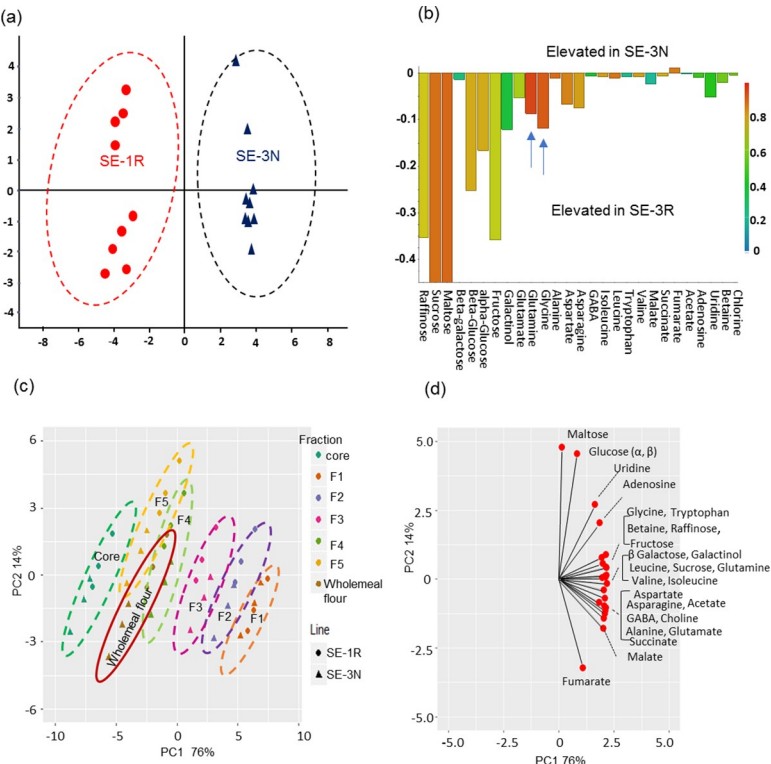

**Fig 4. Metabolite data analysis of RNAi and null lines by $^1$H-NMR.** (a), OPLS-DA of polar metabolite profiles of wholemeal flour in RNAi line SE-1R and null SE-3N by $^1$H-NMR. (b), Contribution plot comparing significant component changes from two lines SE-1R and SE-3N. The red colour represents more elevated in SE-3N on the top, and in SE-1R at the bottom. The amino acids (glutamine and glycine) were most elevated in SE-1R, and are indicated by blue arrows. (c), PCA scores plot of pearling fractions: F1, F2, F3, F4, F5, core and wholemeal flour of SE-1R and SE-3N. (d), The loading plot of PCA for pearling fractions. The coloured circles were drawn by separation of different lines in (a) and different fractions in (c), rather than statistically significant. F1, F2, F3, F4, F5 and core are mainly enriched in bran (F1), aleurone cells (F2), sub-aleurone cells (F3), two progressive inner endosperm fractions (F4, F5) and core.

the concentrations of most of the amino acids transported by *TaAAP13* were elevated by the suppression of *TaAAP13* in line SE-1R.

## Overexpression of *TaAAP13* increases grain nitrogen content and grain size

To determine whether overexpression of *TaAAP13* can increase nitrogen accumulation in wheat grain, transgenic plants were generated using *TaAAP13D* driven by two promoters: the starchy endosperm specific wheat high molecular weight glutenin subunit (HMW-GS) 1Dx5 promoter and the constitutive maize ubiquitin promoter. Expression under the control of the ubiquitin promoter did not affect grain nitrogen concentration (S6A Fig). By contrast, expression under the control of the HMW-GS promoter significantly increased grain nitrogen concentration, grain size, thousand grain weight, and nitrogen content per grain (Fig 5).

With the HMW-GS 1Dx5 promoter, the *TaAAP13* expression levels in 21DPA caryopses were increased by 9–12 fold in the transgenic lines P16-OE and P22-OE (containing 6 copies) and by 30–50 fold in the lines P23-OE (28 copies), P24-OE, P25-OE and P26-OE (12–16 copies) compared with a null line (P15-null) and a non-transgenic line (Cadenza, P29-WT) (Fig 5A). The six over-expression lines all had increased concentrations of grain nitrogen compared to the null and control lines, with statistically significant increases of 14.4% to 32.4% in P23-OE, P24-OE and P26-OE (Fig 5B). The thousand grain weight (TGW), N content per grain and grain size were also significantly increased by 19.3–31.7%, 31.2–72.3% and 9.3–34.7% (P<0.05, F-test), respectively, in P16-OE, P23-OE, P24-OE, and P26-OE (Fig 5C and 5D, S4 Table). However, the grain numbers per plant, grain yields per plant and plant biomass were significantly decreased by 60.5 and 71.0%, 50.8 and 62.2%, 40.9% and 38.6% (P<0.05, F-test) respectively, in the overexpression lines (P23-OE and P24-OE) (S6B–S6D Fig).

The grains were longer and wider, but more wrinkled in the overexpression lines with higher transgene copy numbers and higher expression of *TaAAP13* (Fig 5E, S4 Table). To determine whether the distribution of protein was altered by overexpression of *TaAAP13*, thin sections of whole mature grains were observed by light microscopy. This showed that some protein bodies were fused to form a larger matrix in the sub-aleurone cells of the dorsal and lobe regions of overexpression line P23-OE compared to P15-null line (Fig 6), indicating that overexpression of *TAAAP13* also affected the distribution of protein in the starchy endosperm. SDS-PAGE showed that overexpression of *TaAAP13* increased ω-gliadins and other gliadins (S7A Fig), with a significant increase in the proportion of ω-gliadins in three lines and a significant decrease in the proportion of HMW subunits in two of the lines (S7B Fig).

Determination of the profiles of polar metabolites from wholemeal flour using $^1$H-NMR spectroscopy showed that the concentrations of 13 of the 15 free amino acids that were determined, including glutamine, proline and aspartic acid which were most abundant, were increased in the overexpression lines (P23-OE and P24-OE), by 1.5 to 2.2 fold compared with the null and wild type lines (S8 Fig). Increases were also observed in the most abundant soluble sugars: glucose, fructose and sucrose. The overexpression of *TaAAP13* therefore supported the results of the yeast complementation, suggesting an ability to transport a broad range of amino acids *in planta*. Partial least squares discriminant analysis (PLS-DA) of 28 metabolites (including 15 amino acids) showed that the first two X-variates (accounting for 72.8% of the explained variation) separated the six overexpression lines from the P15-null line and the non-transgenic P29-WT line (Cadenza) (S9A Fig). In particular, the control lines are positively associated with X-variate 1 and negatively associated with X-variate 2. These X-variates are defined by their loadings, shown in S9B Fig. The results indicate that overexpression of *TaAAP13* also changed the profiles of metabolites in the grains by accumulation of more free amino acids and soluble sugars in transgenic lines compared to the null or wild type lines.

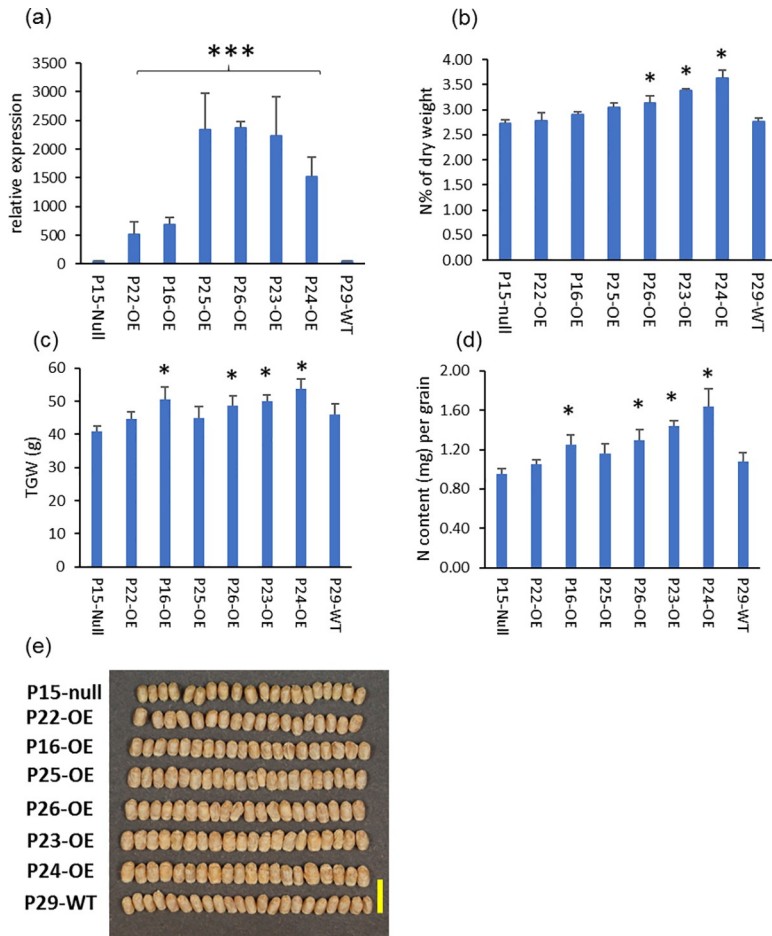

**Fig 5. The gene expression and phenotype of overexpression *TaAAP13D* under the control of endosperm–specific promoter HMW-GS 1Dx5.** (a), Gene expression of *TaAAP13* at 21DPA in developing grains of overexpression (OE) lines: P22-OE and P16-OE (6 copies), P25-OE (16 copies), P26-OE (14copies), P23-OE (26 copies), P24 (12 copies), null (P15-null), and non-transgenic (P29-WT, Cadenza). Significant differences between OE lines and null or wild type were detected ($F_{2,14}$ = 20.85, p<0.001) (P<0.001, F-test) and indicated with asterisk (\*\*\*). (b), N concentration % of wholemeal flour. (c), Thousand grain weight (TGW) at 15% moisture content (g). (d), Nitrogen content (mg) per grain. (e), Grain morphology of T3 grains. Scale bar represents 1 cm. Significant differences between OE lines and null were detected using ANOVA (P<0.05, F-test) and indicated with asterisk (\*).

## Discussion

Nutrients (amino acids, sucrose, and monosaccharides) are transported from the vascular bundle of the grain in the crease into transfer cells in the nucellar projection, where they are released into the endosperm cavity [30], and subsequently taken up by the endosperm transfer cells (as shown schematically in Fig 7). The endosperm transfer cells are highly specialized cells with secondary wall ingrowth, which can amplify the membrane area up to 20-fold at 25 DPA and consequently enhance the efficiency and capacity of transport of solutes [51, 52]. In this study, *TaAAP2* was highly expressed in the endosperm transfer cells (ETC) during grain filling (14–28 DPA), which is associated with increased ingrowth of the endosperm transfer cell walls [53] and increased protein synthesis in the starchy endosperm [54]. Yeast complementation confirmed that *TaAAP2* could transport a broad range of neutral amino acids, which was consistent with previous results [3, 43, 55, 56]. In addition, most AAPs are located in the plasma membrane [57] and energized by H$^+$ symporter [58]. Therefore, *TaAAP2*

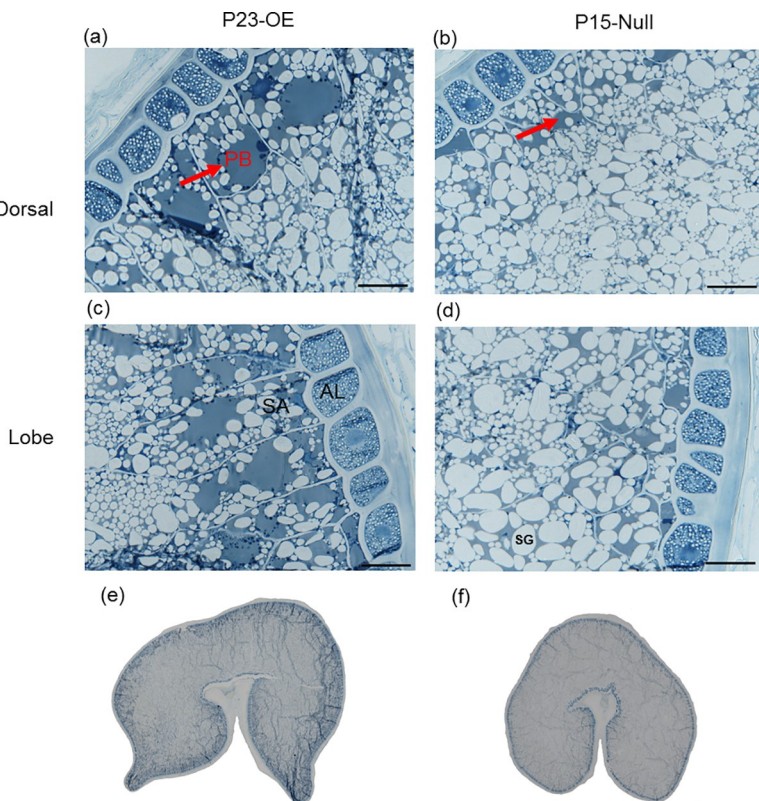

**Fig 6. The protein distribution of mature grain resin sections of *TaAAP13* overexpression line.** (a), Dorsal of P23-OE. (b), Dorsal of P15-null. (c), Lobe of P23-OE. (d), Lobe of P15-null. (e), Whole grain section of P23-OE. (f), Whole grain section of P15-null. Resin sections with 1 μm thickness in (a)-(d) and 0.5μm thickness in (e) and (f) were stained with 1% (w/v) Naphthol Blue Black in 7% (w/v) acetic acid. Scale bars represent 50 μm in (a)-(d) and 1mm in (e) and (f). Red arrows indicated large protein body matrix (PB). AL: aleurone cells, SA: sub-aleurone cells, SG: starch granules.

expression may be coordinated with increased membrane surface area in the endosperm transfer cells to regulate the amino acid uptake rate to meet the demand of grain protein synthesis (Fig 7).

The embryo is isolated symplastically from the endosperm, and the epithelium cells of the scutellum function as transfer cells [59] that can transport amino acids from the apoplast into the embryo (Fig 7B). The aleurone layer has high concentrations of vitamins, minerals, proteins and lipids [60]. The differentiation of aleurone cells is initiated at 6–8 DPA and the accumulation of proteins and minerals occurs between about 11–27 DPA [61, 62]. *TaAAP21* was most strongly expressed in the aleurone and scutellar epithelium between 14–28 DPA, and can complement many amino acids in yeast mutants, implying a role in transport of amino acids into the embryo and aleurone (Fig 7). This is supported by data on the *Arabidopsis* ortholog, *AtAAP1*, which is expressed in the epidermal transfer cells of the embryo and endosperm, thereby facilitating the import of amino acids into the embryo [13]. However, the functions of *TaAAP2* and *TaAAP21* in transporting amino acids in the wheat grain need to be confirmed by direct functional analysis.

Amino acids are transported into the starchy endosperm for storage protein synthesis, and endosperm protein content is dependent on amino acid availability. *TaAAP13* was expressed in the lobes of the starchy endosperm during early endosperm development, expressed more strongly in the sub-aleurone and inner starchy endosperm during the middle phase of grain

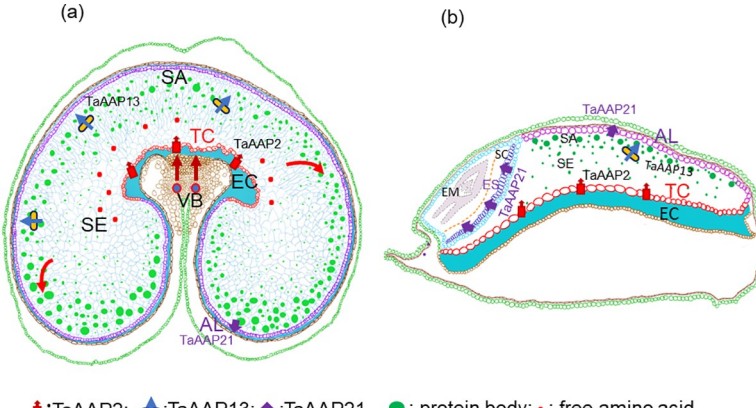

**Fig 7. Schematic view of three amino acid transporters in nitrogen transport.** (a), Wheat grain transverse section. (b), Wheat grain longitudinal section. VB: vascular bundle; EC: endosperm cavity for amino acids delivered to endosperm from vascular bundle; NPTC: nucellar projection transfer cells; ETC: endosperm transfer cells, *TaAAP2* expression localization for uptake of amino acids from endosperm cavity; SE: starch endosperm; SA: sub-aleurone cells for proteins (mainly ω and α-gliadins, LMW-glutenin subunits), and *TaAAP13* expression localization; AL: aleurone cells for *TaAAP21* expression localization. ES: epithelium of scutellum for *TaAAP21* expression localization. SC: scutellum. EM: embryo.

development, but restricted to the sub-aleurone cells during late grain development. The highest level of expression of *TaAAP13* in wheat endosperm was at 14 DPA at the start of the grain filling period and about a week before the maximum rate of protein accumulation (at about 21 DPA) [45, 63, 64]. ω-gliadins, α-gliadins, and low molecular weight subunits of glutenin are concentrated in the sub-aleurone cells and show a gradient from the outer cell layers to the inner endosperm determined by RNA *in-situ* hybridization, promoter::GUS and protein immunofluorescence [45, 65, 66]. *TaAAP13* showed similar expression patterns and localization to storage proteins.

*TaAAP13* can transport 14 amino acids by yeast complementation, notably glutamine, which comprises 50–60% of the free amino acids in the wheat caryopses at 7 DPA [67] and is present in the endosperm cavity (43% of the free amino acid pool) at 21 DPA [68]. In addition, glutamine accounts for 30 and 50% of wheat amino acid residues in the gluten proteins stored in the wheat starchy endosperm [69]. Therefore, the glutamine availability is important for gluten protein synthesis.

Suppression of *TaAAP13* expression did not reduce the total concentration of nitrogen in the mature grain or the radial distribution shown by pearling, but did result in increased concentrations of free amino acids (especially glutamine, glycine and asparagine) in both wholemeal flour and pearling fractions. A similar effect on total free amino acids was also observed in *AtAAP1*, *UMAMIT11*, *14, 28* and *29* loss-of-function mutants of *Arabidopsis* seeds [13, 17]. The concentrations of amino acids were also increased by more in the inner starchy endosperm and core fractions produced by pearling.

The amino acids taken up by endosperm transfer cells are transported across the starchy endosperm tissue to the sub-aleurone cells via a coordination of symplastic and apoplastic routes [52, 58]. However, there is no direct evidence to prove the cellular pathway of amino acid transport in wheat starchy endosperm. Ugalde and Jenner (1990) reported that the amino acid concentrations in the dorsal endosperm showed a decrease from the endosperm cavity to the midpoint due to the influence of high concentrated fluid in endosperm cavity, followed by an increase from the midpoint to the periphery [70]. However, in this study, the concentrations of 11 free amino acids increased from the endosperm core to the aleurone cells (present

in pearling fraction F2) in both the RNAi and null lines (S3A Table), indicating that amino acid transporters are required to transport nitrogen to the outer endosperm against gradients in amino acid concentrations. Most known *Arabidopsis* AAPs are active, proton-coupled amino acid symporters [56, 71, 72]. We therefore suggest that suppression of *TaAAP13* expression reduced the import of the free amino acids (glutamine, glycine, and asparagine) into the sub-aleurone cells, which subsequently resulted in greater accumulation of free amino acids in the inner endosperm cells in the RNAi line compared to the null line (Fig 7). However, the concentrations of amino acids were also increased in mature grains of lines with overexpression of *TaAAP13* under the control of HMW-GS. This suggests that amino acid transport capacity was increased by overexpressing *TaAAP13*, with more total nitrogen accumulation in individual grains. Free amino acids were increased both in RNAi and overexpression lines of *TaAAP13*, but the patterns of accumulation are different with the free amino acid distribution being altered in RNAi grain and more free amino acids being imported into overexpressing grain.

The overexpression of *TaAAP13* not only increased the nitrogen concentration and the nitrogen content per grain, but also increased the concentration of gluten proteins, particularly ω-gliadins. These results suggest that *TaAAP13* may play a role in importing amino acids into the endosperm for storage protein synthesis during the middle and late grain filling stages. The ectopic expression of *HvSUT* in wheat endosperm and *VfAAP1* in pea cotyledons resulted in increased accumulation of gliadins and globulins in seeds respectively [18, 73]. The greater effect of *TaAAAP13* on ω-gliadins is consistent with the ω-gliadins being highly responsive to the nitrogen status [45].

The higher thousand grain weight (TGW) and larger grain size in the OE lines resulted mainly from the increased grain protein content. The remobilization and translocation of nitrogen to spikes strongly affects grain number [24] and TGW and grain size showed both negative relationship with grain number per plant in the OE lines. This suggests that the increased sink capacity of the OE grain resulted in the redistribution of the limited nutrient supply into a smaller number of larger grains. Previous studies showed that TGW and grain size were greatly increased by ectopic expression of barley sucrose transporter (*HvSUT*) in wheat grain [73] and of *VfAAP1* in peas [18], with grain mumber per spike being decreased in wheat. Grain size is associated with the numbers and size of cells in the grain. Expressing the *Arabidopsis* phloem-specific sucrose transporter (*AtSUC2*) in rice phloem increased grain size, which resulted mainly from an increase in cell size and not cell number for large endosperm [74]. In *TaAAP13* OE grains, more large cells filled with protein matrix were observed in the sub-aleurone, which may have contributed to the increased grain size. Unexpectedly, the plant biomass was lower in the OE lines, and it is possible that the strong sink for nitrogen in the endosperm could have affected embryo development by reducing the nutrient supply, which consequently had a negative impact on plant biomass. In contrast with the OE lines, no effects of *TaAAP13* on TGW, total grain nitrogen content or grain size were observed in the RNAi lines. This may be because the RNAi suppressed *TaAAP13* expression in whole plants, not only in the grain. However, more work is required to further investigate the mechanisms determining increased grain size, nitrogen accumulation, and their impact on plant biomass of OE lines in the future.

The study therefore suggests that overexpression of *TaAAP13* in the starchy endosperm increased sink strength and hence grain size and weight by increasing the nitrogen uptake capacity of the grain. However, this resulted in reduced grain number due to limited availability of assimilate. This suggests that increases in grain nitrogen content and grain yield would require simultaneous increases in remobilized nitrogen to spike and in importing nitrogen in grain for future wheat breeding

## Supporting information

**S1 Fig. The expression patterns of *TaAAP2*, *TaAAP13*, and *TaAAP21* in wheat organs by RNA-seq.** The *TaAAP2*, *TaAAP13*, and *TaAAP21* expression patterns in different organs of Chinese Spring by RNA-seq at leaf, root, stem, spike and grains (Z71, 2DPA, Z75, 14DA, Z85, 30DPA). The expression unit was expressed as FPKM (frequency per kilobase million).
(TIF)

**S2 Fig. The localization of *TaAAP2B*, *TaAAP13D*, and *TaAAP21A* expression by promoter:: GUS analysis.** Grain GUS staining from transgenic cadenza *TaAAP13D* in (a-b), and *TaAAP21A* (c-f) at grain 21DPA (a, c, d, e), and at 28DPA (b, f). Root Gus staining from germinated grain (2–3 days) in *TaAAP2B* (g), *TaAAP13D* (h), and *TaAAP21A* (i). Negative control of Cadenza (j-l) at 14DPA (j), 21DPA(k), and 28DPA(l). The images were visualized using fresh grain (a, b, j, k) or after serial ethanol de-staining (c-i, l). Scale bars represent 750μm (b, l), 500μm (c, g, f, j, k), and 250μm (a, h, g, h, I). AL: Aleurone; EM: embryo; SA: sub-aleurone cells.
(TIF)

**S3 Fig. Complementation of yeast mutants by *TaAAP2*, *TaAAP13*, and *TaAAP21*.** The yeast transformants with *TaAAP2*, *TaAAP13*, *TaAAP21*, *AtAAP2* (positive control) and yeast with empty vector pDR196 (negative control) were grown on selective medium containing 1mM proline or citrulline respectively.
(TIF)

**S4 Fig. The agronomic traits of the RNAi lines (SE-1R, SE-9R, SE-10R), Nulls (SE-3N, SE-11N), and non-transgenic line (SE-24WT).** (a), Grain number per plant. (b), 1000 grain weight (g) at 15% moisture. (c), Biomass for above ground vegetative tissue (g) per plant. (d), Grain yield (g) per plant. (e), Grain areas. There is no statistically significant difference between transgenic lines and nulls except SE-10R and SE-11N in grain area. Significant differences were detected using ANOVA ($P < 0.05$, F-test).
(TIF)

**S5 Fig. The nitrogen concentration and free amino acids in RNAi lines of whole grain flour and pearling fractions.** (a), Nitrogen concentration in pearling fractions (F1, F2, F3, F4, F5 and core) of the RNAi lines (SE-1R, SE-10R), Null (SE-3N), and non-transgenic line (SE-24WT). There is no statistically significant difference between SE-1 and SE-3 ($P < 0.05$, F test). (b), Total free amino acid content in whole grain flour of RNAi lines (SE-1R, SE-9R, SE-10R), null lines (SE-3N, SE-SE11N) and non-transgenic line (SE-24WT, cadenza) determined by HPLC. (c), The metabolite changes of pearling fractions and whole grain flour by $^1$H-NMR in SE-1R (RNAi) and SE-3N (Null). The data on the outside and inside Y axis represent log and original amino acid concentrations (mg/g dry weight), respectively. F1, F2, F3, F4, F5 and core are mainly enriched in bran (F1), aleurone (F2), sub-aleurone (F3), towards inner endosperm (F4, F5 and core).
(TIF)

**S6 Fig. Phenotypes of overexpression of *TaAAP13* lines.** (a), wholemeal flour nitrogen concentration of overexpression of *TaAAP13* lines under maize promoter Ubiquitin. (b)-(f), Overexpression of *TaAAP13* lines under the control of wheat HMW-GS 1Dx5 promoter. (b), Seed number per plant. (c), Grain yield (g) per plant. (d), Biomass (g) per plant. Significant differences were detected using ANOVA ($P < 0.05$, F-test) and were indicated with asterisk (*).
(TIF)

**S7 Fig. SDS-PAGE and protein composition of overexpression lines.** (a), SDS-PAGE of total protein, lane1: P-15null, 2: P23-OE, 3: P24-OE, 4: P25-OE. The image was spliced together

from the same original gel (in S10 Fig. P15-R1, P23-R1, P24-R2, and P25-R3 respectively). (b), The protein composition of HMW-GS, ω-gliadin, LMW-GS, and other gliadins. Percentage (%) is each protein group as a % of total glutens. Significant differences were detected using ANOVA (P<0.05, F-test) and were indicated with asterisk (*).
(TIF)

**S8 Fig. Metabolites of wholemeal flour from overexpression lines were determined by[1]H-NMR.** a), Free amino acids. (b-c), Other metabolites. Significant differences were detected using ANOVA (P<0.05, F-test) and were indicated with asterisk (*).
(TIF)

**S9 Fig. A partial least squares discriminant analysis (PLS-DA) of metabolites from overexpression lines by [1]H-NMR.** (a), PLS-DA plot. P15-null and P29-WT as control compared with all six overexpression lines (OE) (P16-OE, P22-OE, P23-OE, P24-OE, P25-OE, and P26-OE). (b), Loading of PLS-DA plot.
(TIF)

**S10 Fig. The original image of overexpression TaAAP13 SDS-PAGE in S7A Fig.**
(TIF)

**S1 Table. Primers for RT-qPCR, promoter cloning, full length CDS cloning of *TaAAP2*, *TaAAP13*, and *TaAAP21* into yeast vector, RNAi and overexpression plasmid vector constructs.** Underlined sequences were added restriction enzyme digestion sites.
(DOCX)

**S2 Table. The amino acids of *TaAAP2*, *TaAAP13*, and *TaAAP21* complementation in yeast mutants.** Cross (×) represents that the AAP is unable to transport this amino acid; Tick (√) represents that the AAP is able to transport this amino acid. † indicates the *TaAAP* is unable to complement yeast mutant at low Arginine concentration 0.5g/l, but is weakly complement yeast mutant at high Arginine concentration at 1g/l.
(DOCX)

**S3 Table. The metabolite changes in RNAi line (SE-1) and Null (SE-3) wholemeal flour and their pearling fractions by 1H-NMR.**
(XLSX)

**S4 Table. The grain size (area), length, and width of pHMW-TaAAP13 overexpression lines.** The data are expressed as means±SE (standard error). Grain size LSD (5%) = 1.72; grain length LSD (5%) = 0.22; grain width LSD (5%) = 0.22. * represents a difference exceeding the LSD 5% level compared with P15 null using ANOVA.
(DOCX)

**S5 Table. Data availability.**
(XLSX)

## Acknowledgments

Yan Wang and Zhiqiang Shi acknowledge the China Scholarship Council for their visit to Rothamsted Research. The authors thank Rothamsted Bioimaging staff for their support.

## Author Contributions

**Conceptualization:** Yongfang Wan, Peter R. Shewry, Malcolm J. Hawkesford.

**Data curation:** Yongfang Wan.

**Formal analysis:** Kirsty Hassall, Stephen Powers.

**Funding acquisition:** Peter R. Shewry, Malcolm J. Hawkesford.

**Investigation:** Yongfang Wan, Yan Wang, Zhiqiang Shi, Doris Rentsch, Jane L. Ward, Caroline A. Sparks, Alison K. Huttly, Peter Buchner.

**Supervision:** Peter R. Shewry, Malcolm J. Hawkesford.

**Writing – original draft:** Yongfang Wan.

**Writing – review & editing:** Yongfang Wan, Doris Rentsch, Peter R. Shewry, Malcolm J. Hawkesford.

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
