## [Decision Letter · Decision Letter 0]

18 Dec 2020

PONE-D-20-34026

Wheat amino acid transporters highly expressed in grain cells regulate amino acid accumulation in grain

PLOS ONE

Dear Dr. Hawkesford,

Thank you for submitting your manuscript to PLOS ONE. After careful consideration, we feel that it has merit but does not fully meet PLOS ONE’s publication criteria as it currently stands. Therefore, we invite you to submit a revised version of the manuscript that addresses the points raised during the review process.

We look forward to receiving your revised manuscript.

Kind regards,

Jin-Song Zhang, Ph.D.

Academic Editor

PLOS ONE

Journal Requirements:

Reviewers' comments:

Reviewer's Responses to Questions

**Comments to the Author**

1. Is the manuscript technically sound, and do the data support the conclusions?

Reviewer #1: Yes

Reviewer #2: Yes

2. Has the statistical analysis been performed appropriately and rigorously? 

Reviewer #1: Yes

Reviewer #2: Yes

3. Have the authors made all data underlying the findings in their manuscript fully available?

Reviewer #1: Yes

Reviewer #2: Yes

4. Is the manuscript presented in an intelligible fashion and written in standard English?

Reviewer #1: No

Reviewer #2: Yes

5. Review Comments to the Author

Reviewer #1: This MS reveals the important role of three amino acid transporters (TaAAP2, TaAAP13, TaAAP21) in wheat grains for amino acid transport, which deserves to be published. Several suggestions are as follows:

1 There are obvious deficiencies in the introduction of this MS. many important AAP genes in Arabidopsis and rice should be introduced, such as AtAAP2 and AtAAP8 in Arabidopsis, OsAAP3, OsAAP5 and OsAAP1 in rice. These genes have an important contribution to the grain yield of Arabidopsis or rice.

2 What is the sub-location of these three amino acid transporters in the plant cell? subcellular localization deserves consideration for supplementation.

3 why are there no overexpression lines and RNAi lines of the other two genes (TaAAP2 and TaAAP21)?

4 The sentences in the MS should be checked and organized, and there are many minor problems.

Reviewer #2: The authors use RNA-seq, RT-qPCR, and promoter-GUS assays to investigate the expression pattern of three amino acid transporters in different wheat tissues. They provide evidence to support that all three transporters can transport a broad spectrum of amino acids. Finally, manipulating the expression of amino acid transporters alters nitrogen accumulation in the wheat grains. The manuscript will be of interest to readers with foci on amino acid transporters as well as those exploring regulation of nitrogen accumulation in the crop grains.

Major comments:

1. The expressions of TaAAP2, TaAAP13, and TaAAP21 in different organs were shown in fig 1d-f. These genes all have three homologous genes, however, the pictures and article only generally show that TaAAP2, TaAAP13, and TaAAP21 are analyzed. Please note that the expression of all their homologous genes or one homologous gene was analyzed.

2. Lines 299-302, in order to determine the expression pattern of TaAAP2, TaAAP13, and TaAAP21, the promoters of TaAAP2b, TaAAP13d and TaAAP21a were selected for GUS staining experiment in Fig. 2. TaAAP2, TaAAP13, and TaAAP21 all have three homologous genes, but why did authors choose the promoter of TaAAP2b, TaAAP13d and TaAAP21a genes? Their high expression or other reasons should be explained.

3. Several yeast mutants were used to examine the transport activities of TaAAP2, TaAAP13, and TaAAP21. The genotypes of these mutants should be indicated or the references for the origin of the mutants should be provided.

4. In this paper, the transport activities of TaAAP2, TaAAP13, and TaAAP21 for several amino acids was examined. Why is histidine not detected?

5. Lines 443-444, which of the TaAAP13 homologous genes is overexpressed should be specified.

Mirror points

1. Line385, "whole grain four" is wrongly written, it should be "whole grain flour".

2. Line407, " flours " should not be plural.

3. Some gene names are not italicized in the paper, such as line260, 283. Please correct it.

6. PLOS authors have the option to publish the peer review history of their article (what does this mean?). If published, this will include your full peer review and any attached files.

Reviewer #1: No

Reviewer #2: No

---

## [Author Response · Author response to Decision Letter 0]

24 Jan 2021

Reviewer #1: This MS reveals the important role of three amino acid transporters (TaAAP2, TaAAP13, TaAAP21) in wheat grains for amino acid transport, which deserves to be published. Several suggestions are as follows:

1.There are obvious deficiencies in the introduction of this MS. many important AAP genes in Arabidopsis and rice should be introduced, such as AtAAP2 and AtAAP8 in Arabidopsis, OsAAP3, OsAAP5 and OsAAP1 in rice. These genes have an important contribution to the grain yield of Arabidopsis or rice.

Answer: We thank the reviewer for this suggestion and have now expanded the introduction to include AtAAP2 in Arabidopsis and OsAAP1, 3, and 5 in rice. See Lines 92-100.

2. What is the sub-location of these three amino acid transporters in the plant cell? subcellular localization deserves consideration for supplementation.

Answer: Most of AAPs are plasma membrane transporters (in Arabidopsis) or ER membrane transporter (OsAAP6). We do not know the sub-cellular localization of these three wheat amino acid transporters, and therefore could not add more data. 

3. why are there no overexpression lines and RNAi lines of the other two genes (TaAAP2 and TaAAP21)?

Answer: TaAAP13 has the gene expression patters, which correspond to the endosperm storage protein deposition, and the grain nitrogen content is dependent on the amino acid availability in endosperm. The TaAAP13 overexpression under the control of HMW-GS promoter has large effect on the grain nitrogen concentration. Therefore, TaAAP13 was selected for investigation of overexpression and RNAi lines. 

4.The sentences in the MS should be checked and organized, and there are many minor problems.

Answer: We carefully checked the manuscript, and corrected the errors. 

Reviewer #2: The authors use RNA-seq, RT-qPCR, and promoter-GUS assays to investigate the expression pattern of three amino acid transporters in different wheat tissues. They provide evidence to support that all three transporters can transport a broad spectrum of amino acids. Finally, manipulating the expression of amino acid transporters alters nitrogen accumulation in the wheat grains. The manuscript will be of interest to readers with foci on amino acid transporters as well as those exploring regulation of nitrogen accumulation in the crop grains.

Major comments:

1. The expressions of TaAAP2, TaAAP13, and TaAAP21 in different organs were shown in fig 1d-f. These genes all have three homologous genes, however, the pictures and article only generally show that TaAAP2, TaAAP13, and TaAAP21 are analyzed. Please note that the expression of all their homologous genes or one homologous gene was analyzed.

Answer: The expression data for TaAAP2, TaAAP13, and TaAAP21 from the qRT-PCR in Fig 1d-f are presented for single homoeologues, as are the promoter GUS analyses (TaAAP2B, TaAAP13D, and TaAAP21A), not the total expression levels of the three homoeologues of each gene. This is stated in Line 315-316 (Fig 1 legend).

2. Lines 299-302, in order to determine the expression pattern of TaAAP2, TaAAP13, and TaAAP21, the promoters of TaAAP2b, TaAAP13d and TaAAP21a were selected for GUS staining experiment in Fig. 2. TaAAP2, TaAAP13, and TaAAP21 all have three homologous genes, but why did authors choose the promoter of TaAAP2b, TaAAP13d and TaAAP21a genes? Their high expression or other reasons should be explained.

Answer: The selection of the three homoeologues for promoter-GUS analysis was based on the gene expression levels and the length of promoter sequence available in the database. This is explained in lines 156-158 (Plasmid vector construction). The genomic sequencing of wheat was not complete when the work was initiated, and sufficiently long promoter sequences were available for limited numbers of genes.

3. Several yeast mutants were used to examine the transport activities of TaAAP2, TaAAP13, and TaAAP21. The genotypes of these mutants should be indicated or the references for the origin of the mutants should be provided.

Answer: The genotypes or references for the mutants of the yeast in amino acid transport assay were 

specified in the method in Line 195-199.

4. In this paper, the transport activities of TaAAP2, TaAAP13, and TaAAP21 for several amino acids was examined. Why is histidine not detected?

Answer: We did not have yeast mutants for all amino acids, including histidine. 

5. Lines 443-444, which of the TaAAP13 homologous genes is overexpressed should be specified.

Answer: TaAAP13D was used for overexpression and this is stated in the Results (line 450) and in the Fig 5 legend (line 457).

Mirror points

1. Line385, "whole grain four" is wrongly written, it should be "whole grain flour".

Answer: corrected.

2. Line407, " flours " should not be plural.

Answer: corrected.

3. Some gene names are not italicized in the paper, such as line260, 283. Please correct it.

Answer: All the gene names are now written in italic.

---

## [Editor Report · Decision Letter 1]

26 Jan 2021

Wheat amino acid transporters highly expressed in grain cells regulate amino acid accumulation in grain

PONE-D-20-34026R1

Dear Dr. Hawkesford,

We’re pleased to inform you that your manuscript has been judged scientifically suitable for publication and will be formally accepted for publication once it meets all outstanding technical requirements.

Kind regards,

Jin-Song Zhang, Ph.D.

Academic Editor

PLOS ONE
---

## [Editor Report · Acceptance letter]

29 Jan 2021

PONE-D-20-34026R1 

Wheat amino acid transporters highly expressed in grain cells regulate amino acid accumulation in grain 

Dear Dr. Hawkesford:

I'm pleased to inform you that your manuscript has been deemed suitable for publication in PLOS ONE. Congratulations! Your manuscript is now with our production department. 

Kind regards, 

on behalf of

Prof. Jin-Song Zhang 

Academic Editor

PLOS ONE